## METHOD

# STRling: a k-mer counting approach that detects short tandem repeat expansions at known and novel loci

Harriet Dashnow[1], Brent S. Pedersen[1,2], Laurel Hiatt[1], Joe Brown[1], Sarah J. Beecroft[3,4], Gianina Ravenscroft[4], Amy J. LaCroix[5], Phillipa Lamont[6], Richard H. Roxburgh[7], Miriam J. Rodrigues[7,8], Mark Davis[9], Heather C. Mefford[5], Nigel G. Laing[4,9] and Aaron R. Quinlan[1*]

*Correspondence:
aquinlan@genetics.utah.edu

[1] Department of Human Genetics, University of Utah, Salt Lake City, UT, USA
[2] Utrecht University Medical Center, Utrecht, The Netherlands
[3] Pawsey Supercomputing Research Centre, Kensington, WA, Australia
[4] Harry Perkins Institute of Medical Research and Centre for Medical Research, University of Western Australia, Perth, WA, Australia
[5] Department of Pediatrics, Division of Genetic Medicine, University of Washington, Seattle, WA 98195, USA
[6] Neurogenetic Unit, Royal Perth Hospital, Perth, WA, Australia
[7] Neurology, Auckland City Hospital, Auckland, New Zealand
[8] Centre for Brain Research, University of Auckland, Auckland, New Zealand
[9] Neurogenetics Unit, Department of Diagnostic Genomics, PathWest Laboratory Medicine, Western Australian Department of Health, Nedlands, Australia

## Abstract

Expansions of short tandem repeats (STRs) cause many rare diseases. Expansion detection is challenging with short-read DNA sequencing data since supporting reads are often mapped incorrectly. Detection is particularly difficult for "novel" STRs, which include new motifs at known loci or STRs absent from the reference genome. We developed STRling to efficiently count k-mers to recover informative reads and call expansions at known and novel STR loci. STRling is sensitive to known STR disease loci, has a low false discovery rate, and resolves novel STR expansions to base-pair position accuracy. It is fast, scalable, open-source, and available at: github.com/quinlan-lab/STRling.

## Background

Short tandem repeats (STRs), are 1–6 bp repetitive DNA sequences that comprise ~3% of the human genome and are highly polymorphic, with mutation rates 10–100,000 times higher than other loci [1]. At least 48 STR expansions cause Mendelian human diseases, such as Huntington's disease and spinocerebellar ataxia (SCA) [2]. Disease mechanisms include polyglutamine aggregation, RNA toxicity, altered methylation, and repeat-associated non-ATG translation [3]. STR variation has also been associated with autism, intelligence, depression, and schizophrenia risk [4–8]. Supporting a mechanistic link, STR variation has been associated with expression levels of nearby genes [9]. Modern DNA sequencing has enabled new software to characterize STR variants at known loci. However, several recently discovered pathogenic STR loci or alleles, including STR expansions implicated in cerebellar ataxia, neuropathy, and vestibular areflexia syndrome (CANVAS); Baratela-Scott syndrome; and several forms of familial adult myoclonic epilepsy (FAME) and SCA (Additional

file 1: Table S1) [10–14], are "novel" in that they include new repeat units at annotated STR loci, or new STR loci where the sequence is completely absent from the reference genome. For example, in CANVAS, the non-pathogenic AAAAG STR found in the reference is replaced by an AAGGG repeat, which, when expanded, causes disease. In Baratela-Scott syndrome, the pathogenic expansion occurs within a 238-bp non-STR insertion relative to the reference genome. Finding a novel STR may indicate that the reference was generated from an individual with an alternate haplotype, or that an error occurred in the assembly of the reference genome.

Typically, researchers aim to discover a disease-causing variant in a single patient, or occasionally a small cohort of individuals with similar symptoms. When presenting with symptoms typically associated with a disease caused by STR variants, patients may have been screened for relevant common STR disease expansions. Genotyping STR expansions using established laboratory techniques such as conventional or repeat-primed polymerase chain reaction (PCR), Southern blot, capillary, or pulse-field gel electrophoresis is expensive and time-consuming, requiring locus-specific assay development. Some phenotypes may be caused by one of several SNV and STR variants, such that even disease panels may still miss causal STR expansions [15]. Increasingly, researchers are moving to genome sequencing, which is often more economical and may yield a faster diagnosis [16]. While long-read sequencing strategies like PacBio and Oxford Nanopore are increasing in popularity, they are still prohibitively expensive for large-scale genomic studies. Additionally, we need methods to analyze the many short-read genomes of patients with unsolved rare diseases.

Several existing methods are capable of genotyping STR alleles shorter than the length of typical Illumina reads, including LobSTR, HipSTR, and RepeatSeq [17–19]. However, the pathogenic allele size for most known STR disease loci exceeds the limits of these methods [3]. More recently, several methods have been developed to detect STR alleles greater than the read length: ExpansionHunter, STRetch, exSTRa, GangSTR, and TREDPARSE [20–25]. While these methods are effective in detecting pathogenic STR expansions at known loci, they all require knowledge of annotated STR loci. Consequently, they are limited to detecting expansions solely of known STRs, missing novel loci.

Another recently developed method, ExpansionHunter Denovo [26], can also detect expansions in novel STRs. ExpansionHunter Denovo claims to predict the position of novel STR expansions to approximately 500–1000 bp accuracy [27]. Rather than estimating allele sizes, it provides STR counts as a proxy for allele size for long alleles only. ExpansionHunter Denovo can perform either case-control or outlier analysis rather than individual-level results, with the user providing controls.

Moving beyond variant discovery to variant prioritization, filtering for variants that are rare in the population has been shown to be a powerful strategy to prioritize pathogenic SNVs and short indel variants in patients [28]. This has led to the use of large population databases such as gnomAD to enhance the analysis of patient genomes [29]. For known pathogenic STR loci it is frequently observed that pathogenic alleles are typically much larger than those found in unaffected individuals [3]. Bringing together these two approaches, outlier analysis enables the discovery and prioritization of loci across the genome with a larger allele in the affected individual compared

with the rest of the population. This approach has been shown to be successful for prioritizing known pathogenic loci using the STRetch algorithm [22].

Here, we introduce STRling, software capable of detecting both novel and reference STR expansions, including pathogenic STR expansions. It calls alleles both within the read length and greater than the read length. It is capable of accurately detecting the genomic position of expansions. It can also quickly discover and jointly call STRs in thousands of individuals, then prioritize alleles that are large outliers in a given individual. STRling is open source and freely available under the MIT license at https://github.com/quinlan-lab/STRling [30].

## Results

### STRling uses k-mers to detect novel and reference STRs

When aligning short-read DNA sequences, reads arising from STR expansions are frequently mismapped or unmapped. Reads containing substantial STR content will tend to map to the position in the reference genome with the longest matching repeat; we define such loci as STR "sinks." These long perfect repeat tracts act as sinks to which STR-containing reads disproportionately map. However, because STRs occur throughout the genome, the longest locus is unlikely to be the one from which that read truly originated. Because of their large edit distance compared to the reference sequence, reads containing novel STR expansions are likely to be left unmapped. This problem is exacerbated for novel STRs; because these loci do not exist in the reference genome, there is no matching sequence to which to align the read. For this reason, STRling uses k-mer counting to find all the reads with substantial STR content. Once these candidate reads are collected, it then uses their well-mapped "mates" to assign them to their correct locus.

STRling uses an aligned BAM or CRAM file as input and scans candidate reads (those that differ from the reference genome, are aligned to known STR regions, or are unmapped) for k-mer content. STRling does not scan reads that align perfectly (i.e., without mismatches, indels, or clipping) to a non-STR region of the reference genome, as these reads are unlikely to contain high STR content. In each candidate read, STRling counts the number of non-overlapping k-mers from two to six bp. Non-overlapping k-mers are better suited to the task of finding tandem (back-to-back) repeats than the overlapping k-mers commonly used in assembly algorithms. This is done by scanning along the read k bp at a time, then counting the number of times each unique k-mer was observed (Fig. 1A). To retain sensitivity in the case of interruptions to the repeat, for example, one or a few bases inserted that would change the phase, STRling creates all possible rotations of each k-mer sequence and stores the minimum rotation. It then calculates the proportion of the read accounted for by each k-mer. STRling chooses the representative k-mer for that read as the one that accounts for the greatest proportion of the read (Fig. 1A). If multiple k-mers cover equal proportions, it chooses the smallest k-mer. If the representative k-mer exceeds a minimum threshold (see the "Methods" section), STRling considers the read to have sufficient STR content to be informative for detecting STR expansions. STRling does the same for soft-clipped portions of reads to find reads that align to the edges of an STR expansion.

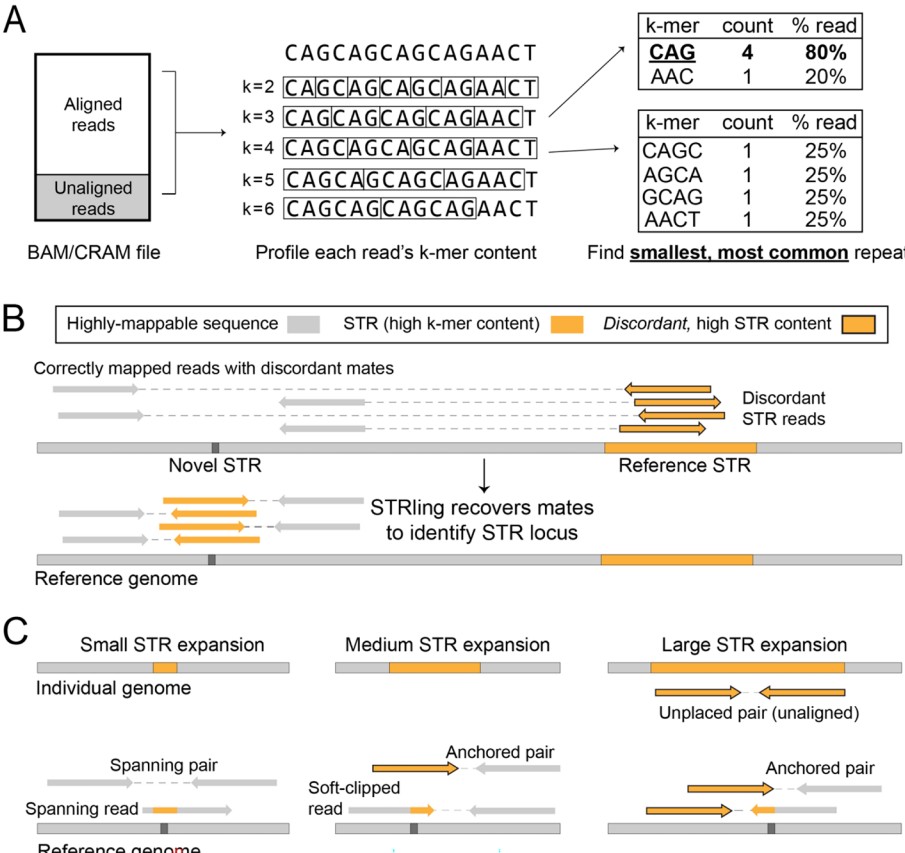

**Fig. 1** STRling uses several types of read evidence to infer STR location and size. **A** STRling performs k-mer counting in reads that are soft-clipped, unaligned, or aligned to a large STR in the reference genome. For each k-mer of length 2–6 bp, STRling selects the one that covers the largest proportion of the read. If two are equal, the smallest is chosen. **B** Where a pair of reads has one read that maps well to the reference genome, and a mate with high STR content, the mapping position of the well-mapped read is used to reposition the STR read. These "anchored pairs" aid in refining the location and improve the quantification of sequence support for the putative STR. **C** Different classes of reads are used to support STR alleles of varying length. Small alleles, shorter than the read length, can be detected by spanning reads, and typically have many spanning pairs. Medium expansions, of a length between the read length and the fragment size, are indicated by anchored pairs and few spanning pairs. Soft-clipped reads can be used to infer the precise insertion point. Large expansions, those longer than the fragment size, are evidenced by a larger number of anchored pairs, as well as contributing unplaced pairs

### Predicting STR expansion loci

For reads with sufficient k-mer STR content, STRling assumes the mapping position to be unreliable and therefore attempts to place the read in its true locus. If the read has a well-mapped non-STR mate then STRling uses the mate's mapping location in conjunction with the sample's median DNA fragment size to relocate the STR read and terms these "anchored pairs" (Fig. 1B). If both reads in the pair have high k-mer STR content, or one is high k-mer and the other is poorly mapped, then the pair is considered unmapped and is recorded as an "unplaced pair" (Fig. 1C).

STRling scans the genome for regions with a cluster of informative anchored and soft-clipped STR reads to identify putative STR expansion sites. Anchored STR reads are used to approximate the "bounds" of the STR expansion, while soft-clipped STR

reads are used to more accurately define the precise insertion point. When performing joint-calling, this procedure is done across informative reads from all samples to produce a joint estimate of the bounds, requiring that at least one sample contribute five reads (by default) for the given bounds to be reported.

Once bounds have been discovered, STRling performs a second, partial pass of the BAM/CRAM to extract additional informative reads for each candidate locus: individual reads that span the bounds ("spanning reads") and pairs of reads that span the bounds, "spanning pairs" (Fig. 1C).

### Estimating allele length

For each individual, STRling uses a combination of spanning reads, anchored pairs, and unplaced pairs to estimate the allele sizes at each locus. From simulations, we have verified that, as expected, the number of anchored pairs is proportional to allele size up to the median fragment length of the sample, while the number of unplaced pairs is proportional to the allele size beyond the median fragment length ("Methods," Additional file 1: Fig. S1). These relationships have been previously described [22]. We therefore used linear models to estimate allele size from these two classes of reads. We used spanning reads to estimate the size of alleles shorter than the read length, if present. If two large expansions (greater than the insert size) with the same repeat unit were present with the same repeat unit there will be a number of pairs for which both reads are completely repetitive. In these cases, we may not be able to identify the source position of the read pair. Some reads would not be assigned; consequently, we would underestimate the size of the allele. However, since many anchored pairs would be present at the edges of the event, we would still correctly identify the position of each expansion.

### Joint-calling and outlier detection

STRling can joint-call large cohorts, allowing the comparison of STR loci across individuals (Fig. 2). Its computational efficiency allows the joint-calling of thousands of samples in parallel. First, STRling collects informative reads for each sample as described above. Then, STRling performs a "joint merge" stage, where candidate STR loci are discovered using reads from all samples. By collecting read evidence across samples, this allows more accurate inference of the STR's boundary in the reference genome. Only those loci with at least five (by default) supporting reads in at least one sample are reported. Allele size estimation is then performed on each sample individually, for each of the loci discovered in the cohort.

For STR diseases with known pathogenic loci and allele sizes, estimating the allele length may be sufficient to detect a likely pathogenic variant. For patients without an expansion in a known disease STR, strategies are needed to prioritize potential new pathogenic variants. Large STR alleles that cause disease are likely to be rare in the general population, and also in patient populations with a mixture of phenotypes. Therefore STRling looks for alleles that are outliers; that is, they are large in a given subject compared with the alleles observed in a set of other genomes. STRling performs outlier analysis across the full cohort and a *z*-score and corresponding *p*-value are generated (see the "Methods" section). These *p*-values are then corrected for multiple testing within each individual. A small *p*-value indicates that an individual harbors an STR expansion

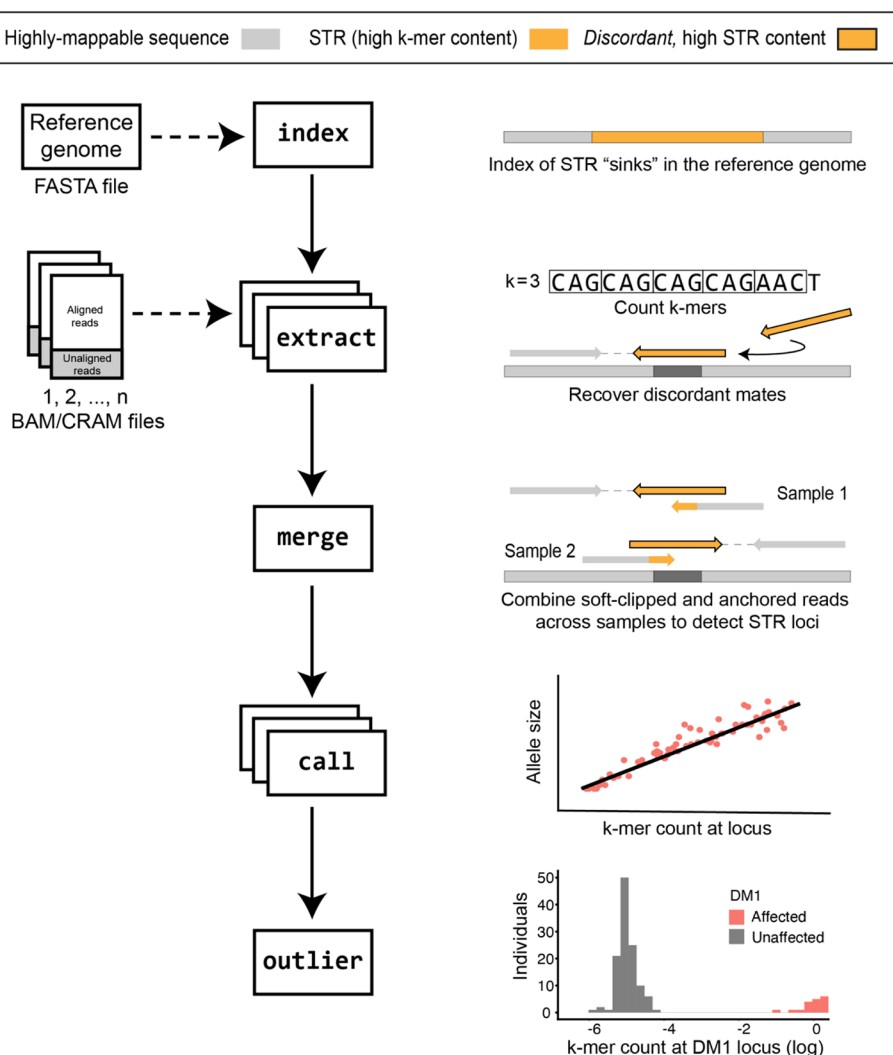

**Fig. 2** STRling joint calling workflow. Index: STRling creates an index of the reference genome, recording the genomic coordinates where large STRs are observed. These regions act as STR "sinks", collecting repetitive reads. Any reads mapping to these regions, in addition to unmapped reads, are candidates to have arisen from a large STR expansion. Extract: STRling counts k-mers to find high STR-content reads, then checks the mate to move the read to its correct position. Merge: read evidence is combined across individuals to increase the accuracy and uniformity of candidate STR expansion loci. Call: STRling estimates the allele sizes using the k-mer count across all reads assigned to a given locus in a linear model. Outlier: STRling checks the distribution across all individuals at a given locus, and tests for outliers

that is rare in the cohort and can be used to prioritize and filter potentially pathogenic STR expansions.

### STRling detects novel and reference STR disease loci

We ran STRling in both individual and joint-calling modes on 134 subjects with whole genome PCR-free, Illumina DNA sequencing. This cohort contains individuals with expansions in 14 known STR disease loci, including 83 affected individuals and 11 carriers (Table 1). For Fragile X Syndrome, there were an additional 17 individuals with premutations, and 22 unaffected family members with alleles in the normal size range. While the majority of the disease STR loci are present in build GRCh38 of the human

**Table 1** Sensitivity of STRling run on PCR-free Illumina WGS of 94 subjects with alleles of pathogenic size at an STR disease locus. Outlier testing was performed against 260 individuals from the 1000 genomes project

| Disease | Inheritance | Repeat unit | CG% | Locus found individual calling | STRling est. > pathogenic threshold | Significant outlier | N subjects |
|---|---|---|---|---|---|---|---|
| **CANVAS** | AR | AAGGG | 60 | 4 (80%) | 0 | 5 (100%) | 5 |
| **DBQD2** | AR | CCG | 100 | 1 (100%) | 1 (100%) | 1 (100%) | 1 |
| DM1 | AD | CAG | 66.7 | 18 (100%) | 18 (100%) | 18 (100%) | 18 |
| DM2 | AD | CCTG | 75 | 1 (100%) | 0 | 1 (100%) | 1 |
| DRPLA | AD | CAG | 66.7 | 2 (100%) | 2 (100%) | 2 (100%) | 2 |
| FRDA | AR | AAG | 33.3 | 26 (100%) | 26 (100%) | 26 (100%) | 26 |
| FTDALS1 | AD | GGGGCC | 100 | 1 (100%) | 0 | 1 (100%) | 1 |
| FXS | XD | CGG | 100 | 11 (68.8%) | 0 | 3 (18.8%) | 16 |
| HD | AD | CAG | 66.7 | 11 (84.6%) | 13 (100%) | 13 (100%) | 13 |
| SBMA | XR | CAG | 66.7 | 1 (33.3%) | 3 (100%) | 3 (100%) | 3 |
| SCA1 | AD | CTG | 66.7 | 3 (75%) | 4 (100%) | 4 (100%) | 4 |
| SCA3 | AD | CTG | 66.7 | 2 (100%) | 2 (100%) | 0 | 2 |
| SCA6 | AD | CAG | 66.7 | 0 | 0 | 0 | 1 |
| SCA8 | AD | CTG | 66.7 | 1 (100%) | 1 (100%) | 1 (100%) | 1 |
| **Total** | | | | **82 (87.2%)** | **70 (74.5%)** | **78 (83.0%)** | **94** |

Novel STR disease loci (not in reference genome) are indicated in bold/underline. Repeat units are reported on the forward strand

*AD* Autosomal dominant, *AR* Autosomal recessive, *XD* X-linked dominant, *XR* X-linked recessive

reference genome, the *CANVAS* pathogenic STRs are new repeat units replacing an annotated STR locus, while the *DBQD2* STR locus is part of a completely novel insertion [11, 12].

When searching for potential pathogenic variants, reasonable filters include the removal of homopolymer expansions (see later discussion of the quality of the variants), limiting the results to autosomes and sex chromosomes, and excluding both low complexity regions (LCRs) and segmental duplications. Using these filters, the 134 subjects tested had a median of 9 (1-252) significant STR expansions each.

Considering those 94 subjects who are affected or carriers for a full mutation, STRling was able to detect the pathogenic STR locus for 82 of 94 (87.23%) subjects using "individual" calling (Table 1). With "joint" calling, known pathogenic loci in 93 of 94 (98.94%) of subjects were detected, while 70 (74.47%) were predicted to be within the pathogenic range based on STRling's predicted allele size. STRling outlier testing identified 83% of pathogenic expansions with an adjusted *p*-value of less than 0.05. The *SCA6* expansion was missed by both individual and joint-calling, yet this is expected, given that the pathogenic allele in this subject is only 26 bp larger than the reference [22]. *SCA6* variants should be able to be found by methods that look for indels within the read. STRling appears to be most sensitive to insertions greater than ~90bp and may struggle to detect smaller pathogenic variants given a single genome. However, STRling was able to correctly identify all pathogenic HD expansions in this cohort (the smallest being 132bp) once joint calling and outlier testing was applied (Additional file 1: Fig. S2A). STRling failed to predict a pathogenic allele size in the *CANVAS, DM2, FTDALS1,* and *FXS* loci. Notably, the pathogenic repeat units for *FTDALS1* (GGGGCC) and *FXS* (CGG) have

100% GC content, and previous methods have described a tendency to underestimate allele size in high-GC content STR loci [20, 22]. With the exclusion of the FXS locus, STRling outlier testing detected over 96% of pathogenic loci. None of the alleles in subjects with verified normal or premutations was predicted to be pathogenic, indicating a low chance of false positives. While STRling does not directly report zygosity, we made an educated guess by inferring that loci with no spanning reads and no or few spanning pairs are likely to be homozygous or hemizygous for a large expansion. When applying this approach to the true positive probands, 55/69 STRling calls had the expected zygosity for the disease mode of inheritance (heterozygous for dominant conditions, homozygous for recessive, hemizygous for X-linked in males). We additionally ran ExpansionHunter Denovo in outlier mode with each single affected individual against the same 260 controls from 1000 Genomes used for STRling. Compared with STRling, ExpansionHunter Denovo (EHdn) had similar sensitivity, except at the FXS locus, where EHdn showed higher sensitivity (Additional file 1: Table S2).

For most known pathogenic loci, STRling was able to identify the genomic position of the expansion to base pair accuracy at most loci (Fig. 3). To quantify how accurately STRling identified the bounds of each locus, we compared the STRling call to the reference positions found in the literature (see the "Methods" section). For individual calling, STRling had a mean position error of 25.3 bp (median: 2, range: 0–241). joint-calling increases locus accuracy by drawing evidence from reads across samples, and greatly reduces the mean position error to 6.14 bp (median: 1, range: 0–155), providing locus resolution that is critical to variant interpretation. The greatest joint-calling

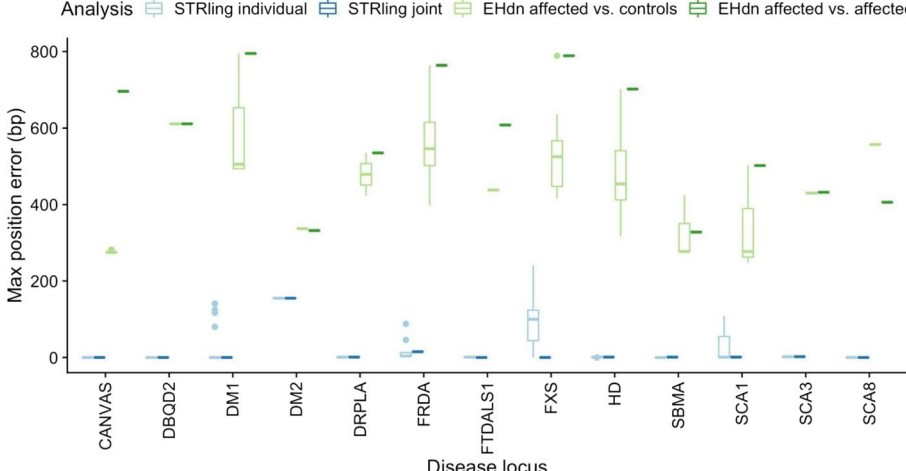

**Fig. 3** STRling shows superior position accuracy at known pathogenic loci. STRling and ExpansionHunter Denovo (EHdn) were run on PCR-free Illumina WGS of 134 subjects with known STR disease status, 94 of which had alleles of pathogenic size (those plotted here). STRling was run on an individual genome "Individual calling" or on all 134 genomes together "Joint calling." EHdn was run with all affected genomes together in outlier mode "EHdn affected vs. affected", or each of the true positives was run in outlier mode with a set of 260 unaffected individuals from 1000 genomes "EHdn affected vs. controls." A locus was considered found if an STR expansion with the pathogenic repeat unit was reported within 500bp of the true locus. Max position error is the position difference between the known and predicted locus (max of upstream and downstream). Zero indicates the predicted position is within or at the bounds of the known locus. STRling was able to detect the true locus position to base pair accuracy for most loci, with greater accuracy using joint-calling, with greater accuracy than ExpansionHunter Denovo under all conditions tested

position error was observed at the DM2 locus, which is a complex locus with the form (TG)n(TCTG)n(CCTG)n, with CCTG expansions associated with disease. We had only one individual in our cohort with DM2, and no soft-clipped read evidence was observed for the CCTG expansion. Therefore, joint calling was not able to improve the position estimate.

As a comparison to STRling, ExpansionHunter Denovo was run in outlier mode with all affected genomes as a single cohort. The resulting mean position error of 713 (median: 764, range: 328–795, Fig. 3) was more than 30 times larger STRling's. STRling demonstrated lower position error for all tested STR loci, likely because, in contrast to ExpansionHunter Denovo, STRling uses the precision of soft-clipped read evidence to improve locus resolution (Figs. 1 and 2). Both STRling and ExpansionHunter Denovo failed to detect the *SCA6* expansion, likely due to its small size (pathogenic expansions are >26bp larger than the reference allele).

For 103 of the subjects with known STR disease, we also had orthogonal allele size estimates from repeat-primed PCR. It should be noted that while PCR-based methods are frequently used in STR disease diagnostics, the accuracy of PCR allele size estimates can suffer from stutter and allelic dropout [31, 32]. When comparing STRling allele size estimates to those from PCR, STRling tends to systematically underestimate allele sizes, especially for larger alleles (Fig. 4). STRling additionally tends to underestimate alleles that are close to the read length of ~150bp. Such underestimates are a consequence of these alleles being in a "gray area" with respect to typical paired-end sequencing protocols: they are too large to be frequently captured by a single read that is typically less than or equal to 150bp, and they are too small to yield a strong anchored pair signal. STRling notably underestimated allele size for the FXS locus, which is a CGG repeat expansion (Additional file 1: Fig. S2A). This locus has been previously identified as

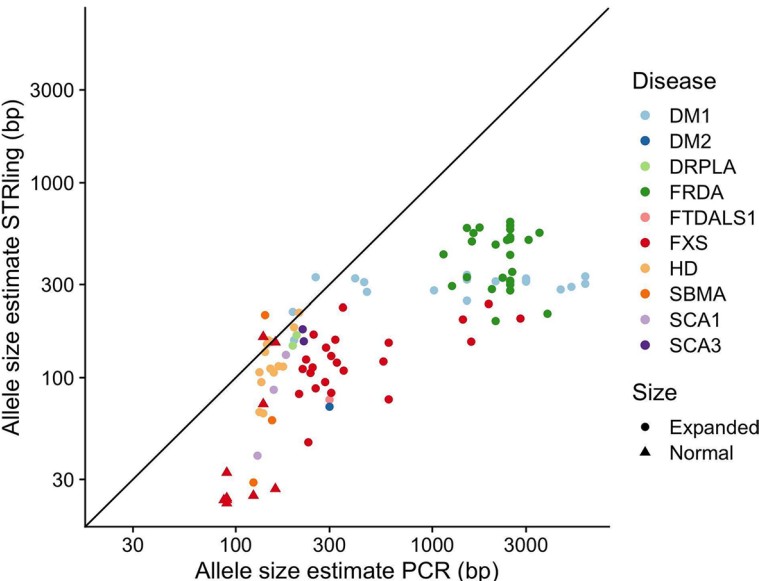

**Fig. 4** Allele size estimates from STRling compared with PCR estimates (log-log scale). STR allele size estimates from 103 individuals also assayed with PCR. "Expanded" includes all pathogenic allele sizes, in both affected individuals and carriers. "Normal" indicates non-pathogenic alleles. The black line indicates *x = y*, equality between STRling and PCR allele size estimates

problematic for Illumina sequencing and allele size estimation, likely due to its 100% GC content [20]. While STRling tends to underestimate large STR expansions, it performs sufficiently well around the pathogenic threshold of most STR disease loci to differentiate pathogenic and non-pathogenic alleles (Table 1).

We joint-called these subjects and performed outlier analysis using an additional set of 260 subjects from the 1000 Genomes Project, a cohort without known STR disease. Each of the true-positive genomes was tested against the set of 260 controls to find outliers. The known pathogenic STR locus was identified as a significant outlier in 83.0% (78 of 94) subjects (Table 1).

In addition, we used this cohort set out to estimate the false discovery rate (FDR) for STRling, assuming that the 260 subjects from the 1000 Genomes Project do not harbor any large pathogenic expansions. We performed joint-calling and outlier testing on the 260 subjects, then filtered the results to known pathogenic STR loci, resulting in 2600 calls in 10 STR loci (Table 2). There were no expansions detected in the other 22 known pathogenic STR loci that we assessed. Of these 2600 calls, 204 were significant, resulting in an estimated FDR of 0.078, yielding a highly-specific average of less than one significant pathogenic expansion per subject. Of the 204 significant calls, only two alleles were estimated to be larger than the pathogenic threshold and most also had evidence of a short, likely non-pathogenic allele (reads and read pairs spanning the locus), suggesting heterozygosity. Note that because many of these diseases are recessive and/or late onset, we would expect some subjects in this cohort to harbor pathogenic STR expansions;

**Table 2** Significant outliers called by STRling in 260 individuals from the 1000 Genomes Project. We performed joint-calling and outlier testing, then filtered the results to 32 well-characterized known pathogenic STR loci (see the "Methods" section). There were 204 significant STRling calls across ten loci (the others had no significant calls), resulting in an estimated FDR of 0.078. Of these, most also had evidence of a short, likely non-pathogenic allele (reads and read pairs spanning the locus), suggesting heterozygosity

| Disease | Significant outlier | Evidence of a short allele | Proportion significant outlier | STRling est. > pathogenic threshold | Inheritance | Notes |
|---------|--------------------|-----------|--------------------|-----------------------|-------------|-------|
| **CANVAS** | 12 | 12 | 0.0462 | 0 | AR | 0.7% carrier frequency [10] |
| **DBQD2** | 5 | 5[a] | 0.0192 | 0 | AR | |
| DM2 | 7 | 7 | 0.0269 | 0 | AD | Age onset: ~30–40 |
| FRA12A | 57 | 57 | 0.219 | 0 | AD | |
| FRAXE | 7 | 7 | 0.0269 | 0 | XR | |
| FRDA | 61 | 61 | 0.235 | 1 | AR | Age onset: 5–25 |
| FTDALS1 | 33 | 32 (1[a]) | 0.127 | 0 | AD | Age onset: 27–85 |
| FXS | 0 | N/A | 0 | 0 | XD | Multiple syndromes with varying pathogenic size thresholds and age of onset: FXS 2, FXTAS 60–65, POI? |
| SCA10 | 8 | 8 | 0.0308 | 0 | AD | Age onset: 12–48 |
| SCA8 | 14 | 14 | 0.0538 | 1 | AD | Age onset: 1–73 |

Novel STR disease loci (not in the reference genome) are indicated in bold/underline

*AD* Autosomal dominant, *AR* Autosomal recessive, *XD* X-linked dominant, *XR* X-linked recessive

[a] No spanning reads, spanning pairs only

therefore the true FDR is likely to be smaller. We found, 4.6% of individuals were significant outliers at the CANVAS locus, which is higher than the previously estimated 0.7% carrier frequency [10]. In this case, the outlier test may be complicated by the fact that only 18% of individuals harbor the haplotype in which the CANVAS expansion arises.

### Long-reads enable estimates of STRling's false discovery rate

In an effort to estimate the number of true and false positive STRling calls outside of known pathogenic STR loci across the genome, we compared STRling calls made based on Illumina sequencing data to STR variants found in long-read HiFi PacBio genome assemblies from the same three individuals sequenced by the Genome in a Bottle consortium: the Ashkenazim trio HG002 (son), HG003 (father), and HG004 (mother) [33]. The original sequencing depth was ~300×, so we subsampled the Ashkenazim trio Illumina sequencing to a more typical ~30× depth to be comparable to other samples. We performed STRling joint-calling of these three individuals in conjunction with 260 controls and tested for outliers. We limited our analysis to STRling calls with a minimum estimated insertion of 20 bp on the canonical chromosomes (chr 1-22, X, Y), and excluded STRling calls that overlapped an annotated segmental duplication, telomere, centromere, or low complexity region (LCR).

We called variants from PacBio HiFi assemblies of the same individuals, filtering to insertions greater than 10 bp. For each STRling call, we selected the closest PacBio call within 500 bp. A STRling call was considered a true positive if the most frequent k-mer in the PacBio insertion matched the STRling repeat unit, or if the STRling repeat unit made up at least 50% of the PacBio insertion. All other STRling calls were considered false positives if they were overlapped by at least one PacBio contig.

Across all three individuals, we observed 1030 true positives and a raw false discovery rate (FDR) of 0.47 across all loci given the above filters (Additional file 1: Table S3). False positives were enriched among homopolymers, with 1 bp repeat units making up 94.6% of all false positives. When excluding homopolymers (i.e., 2–6 bp repeat units), the FDR is reduced to 0.17. Furthermore, for significant outlier STR loci of all repeat units with adjusted *p*-values less than 0.05, the FDR was 0.48, and 0.16 for significant 2–6 bp loci. Once restricted to 2–6 bp loci, limiting to outliers did not substantially reduce the aggregate FDR. However, given the small numbers (10–12 false positives per individual), we expect that increasing the sample size may reveal a lower number of false positives in the outliers. Estimated FDR did not vary substantially across allele sizes (Additional file 1: Fig. S2B).

As *de novo* STR expansions in a proband are prioritized in studies of rare human disease, we further examined the Ashkenazim trio for non-homopolymer *de novo* variants. For each of the 56 significant outlier expansions predicted by STRling in the child that had been confirmed by PacBio and had calls in both parents, we tested for Mendelian concordance or violation in the parents and then checked for supporting evidence in the long-read contigs. For 21 (38%) of these loci, there were one or more missing PacBio contigs in one of the parents that prevented us from validating the *de novo* call. For the remaining 35 loci, 66% (21/32) of the expanded loci that were concordant between the child and both parents were supported by the long-read assemblies. In contrast, none of the 14 apparent *de novo* variants called by STRling were corroborated by PacBio; long-read evidence suggested that

they were inherited. The main cause of invalidation of STRling *de novo* calls was the presence of an expansion in one or both of the parents that were found by PacBio but missed by STRling. This indicates that while overall STRling calls were more often concordant between parent and child, STRling *de novo* calls were highly enriched for errors. While the FDR for apparent *de novo* changes was high, only 69 significant non-homopolymer expansions were called in the child, demonstrating a low genome-wide false positive rate. Therefore, when searching for a pathogenic variant in the context of rare disease, STRling reports a relatively small number of candidate variants.

### Scaling up: locus discovery and algorithm resource requirements

We ran a STRling joint calling on 1000 high-coverage WGS from the 1000 Genomes project. For 2–6 bp repeat unit loci with variants at least 50 bp larger than the reference, we observed expansions in 1549 known and 650 novel STR loci. To explore the impact of sample size on joint calling, we randomly ordered then sampled genomes, recording the number of distinct STR loci that were variable. As sample size increased, so did the number of variable STR loci discovered, with far more reference than novel loci at all sample sizes (Additional file 1: Fig. S4A). There is some indication in the plot of an approaching plateau, however, that has not yet been reached in the sample size presented. We additionally explored the effect of cohort size on outlier testing. We performed the full joint-calling workflow on 20, 100, 200, 500, and 1000 genomes. Increasing cohort size did increase the number of significant outliers reported per individual. After filtering to 2-6 bp repeat units in outside highly repetitive regions (LCRs, segmental duplications, centromeres, and telomeres) this amounted to an average of 10.2 significant outliers in a cohort of 20 compared with 33.6 in a cohort of 1000 (Additional file 1: Fig. S4B).

We ran STRling joint-calling on the 260-individual subset of the 1000 Genomes Project samples used as controls above. The analysis took 373 CPU hours, for an average of 1.43 h per sample (Additional file 1: Fig. S5A). The longest task was the "extract" stage, which finds informative reads and counts k-mers in them (mean: 64.1 min). The max RAM usage occurred during the joint merge stage (29.04 GB) and the joint outlier stage (27.05 GB, Additional file 1: Fig. S5B). Across stages that run on a single sample, the max RAM usage was 2.047 GB. In contrast, the ExpansionHunter Denovo "profile" stage (analogous to STRling's "extract" stage) typically requires less than 40 min and less than 1 GB RAM (Additional file 1: Fig. S6A-B). While STRling requires higher resource requirements than EHdn, STRling provides greater position accuracy and allele size estimates, both of which require assessing additional reads.

STRling's most resource-intensive stage, "merge," is also dependent on the number of samples, scaling at approximately 0.1 GB per genome (Additional file 1: Fig. S5C-D). Additionally, as more individuals are tested, an increasing number of variant STR loci are discovered (Additional file 1: Fig. S5E). This number does appear to begin to plateau at thousands of individuals.

### Discussion

STRling has the potential to go beyond the diagnosis of known STR disease, to discover new STR loci from existing short-read sequencing data with positional accuracy. At pathogenic loci, STRling has high sensitivity (83% for the outlier test), similar

to ExpansionHunter Denovo, except for the Fragile-X Syndrome locus. Critically, STRling was typically able to identify the genomic coordinates of STR expansions to base pair accuracy, in contrast to ExpansionHunter Denovo, which was typically off by more than 700 bp. Position accuracy is vital to interpretation and validation, especially for novel STR loci that have not previously been identified.

While we estimate STRling's false discovery rate to be 0.078 for pathogenic STR loci, some of these individuals may be carriers of recessive alleles or are below the typical age of symptom onset. Therefore, the true rate at which it detects expanded alleles may be lower. For non-pathogenic 2–6 bp STR expansions, STRling's FDR after recommended filters was 0.17, which is similar to the best-performing SV callers [34]. STRling showed limited accuracy for estimating the size of the STR allele, especially for alleles exceeding the insert size. This is a limitation of using short-read sequencing data to detect events larger than the read length.

Other than ExpansionHunter Denovo, STR-calling methods capable of calling large expansions, such as Expansion Hunter, gangSTR, and STRetch, only call reference STRs [21, 22, 24]. This is a key advantage of STRling. It does not rely on the repeat unit being present in the reference genome, and therefore is of particular value where individuals carry haplotypes that differ from the alleles found in the current reference genome. We found 650 variable novel loci in a subset of the 1000 Genomes, accounting for ~30% of expansions >50 bp in that cohort. While the T2T-CHM13 reference may contain many STR loci missing from GRCh38, repeat units often differ between individuals at the same locus (e.g., CANVAS), so having a complete reference is not sufficient to measure all novel STRs [35, 36].

STRling has been developed to scale to thousands of samples. It typically runs in less than 2 h per genome with less than 30 GB peak memory usage for joint stages on 260 samples (scaling at ~0.1 GB per genome) and ~2 GB for individual stages. We provide workflows in three languages, Nextflow [37], Bpipe [38], and WDL [39], for compatibility with cloud services such as AWS, Google Cloud, and Terra (https://strling.readthedocs.io/en/latest/workflows.html).

Although we have used PacBio long reads as an orthogonal truth set to judge the accuracy of STRling STR detection, there are limitations to this approach. Despite the enormous promise of long-read sequencing technologies for their ability to span repetitive sequences, it has been observed that the number of repeat units can vary substantially between long reads in the same individual at the same locus [40]. The implication is that it may be difficult to determine the true allele size of an STR, even with long reads.

Furthermore, while there are key advantages to using long reads to detect STRs, thousands of individuals with rare diseases have been sequenced with short-read technologies and remain without a genetic diagnosis. It is likely that a substantial proportion of patients without a molecular diagnosis may be harboring pathogenic repeat expansions that are evading detection. This is particularly likely to be the case for rare genetic neurodegenerative disease, where STR expansions are a common known cause. Given its accuracy and locus specificity, STRling has the potential to contribute to solving the roughly half of sequenced rare disease cases that remain unsolved and to deepen our understanding of how STRs vary in the wider population [41].

## Conclusions

STRling is a fast and accurate method to detect STR expansions from short-read sequencing data. Critically, it can detect novel expansions, those that are missing from the reference genome. Several such loci are known to cause human disease. In contrast to previous computational methods to detect novel STRs, STRling is capable of defining the locus boundaries to base-pair accuracy. STRling is open source and freely available at https://github.com/quinlan-lab/STRling.

## Methods

### STRling algorithm

STRling is open source and freely available under an MIT license at https://github.com/quinlan-lab/STRling [30]. STRling is predominantly written in the compiled nim language using the hts-nim library [42] with outlier analysis written in Python using pyranges [43], peddy [44], pandas [45], statsmodels [46], numpy [47], and scipy v1.2 [48]. Workflows are available in Nextflow [37], Bpipe [38], and WDL [39] (https://strling.readthedocs.io/en/latest/workflows.html).

#### *Identifying and localizing informative reads with k-mer counting*

Illumina reads containing substantial repetitive content are frequently mis-mapped or left unmapped by alignment algorithms [49]. The first task is to recover STR-containing reads and determine their most likely genomic origin. STRling first makes an index of large uninterrupted STR loci in the reference genome, by searching for perfect repeats in 100 bp windows, sliding by 60 bp at a time. Reads aligning to these STR "sink" regions are considered to have unreliable mapping locations. In the extract subcommand, STRling performs k-mer counting to identify STR content on reads aligning to STR regions, reads that are unmapped or have low mapping quality (MAPQ less than 40), and in the portions of reads that are soft-clipped. STRling counts k-mers of size 2–6 bp, sliding by k to generate non-overlapping k-mers. For each k-mer we perform all possible rotations and store the minimum. This enables us to retain sensitivity in the case of interruptions to the repeat that would change the phase. K-mers are represented as integers to avoid string comparisons, thereby increasing speed. A read is considered informative if any k-mer makes up at least 80% of the sequence by default. The most frequent k-mer is taken to be the representative repeat unit of that read. If that k-mer is a homopolymer (e.g., AA) then the repeat unit is reported to be a single base pair.

 We consider the alignment location of high STR content reads unreliable. Therefore, STRling uses paired information to localize these reads where possible. Specifically, if a read contains at least 80% STR content and has a well-mapped mate (MAPQ > 40 and non-repetitive), STRling uses the mapping position of the mate to relocate the STR read to a position that is the median fragment length away in the direction concordant with the orientation of the mate. This is called an "anchored pair" (Fig. 1). The empirical fragment length distribution is determined for each sample individually. If the mate does not have sufficient mapping quality to be used to relocate the

STR read, then the STR read is considered unmapped. If both reads exceed the STR content threshold then the pair are both considered unmapped and are recorded as an "unplaced pair" (Fig. 1).

### Identifying STR loci

STRling bins informative STR reads by their representative k-mer then scans across them in genomic order looking for clusters of informative reads that may indicate an STR locus. To be considered, a position must have, by default, at least five informative reads, with at least one of those being anchored reads. The median center position of the reads is calculated; any reads that are more than the 98th percentile of the fragment distribution + 100 bp away from the center are removed, and any new reads within range are added. The process is repeated until the position stabilizes or a left clip is discovered, indicating a distinct STR locus. The "left" and "right" bounds of the putative locus are estimated using the edges of soft-clips, or if none are present, the center of the anchored reads.

For joint-calling, locus discovery is performed using reads from all samples using the merge subcommand, and the resulting positions are provided to the call subcommand for individual genotyping. By default, STRling discards any loci where less than five of these reads come from the same sample. The call subcommand additionally performs locus detection on any reads that cannot be assigned to the provided loci so that expansions present in an individual genome can still be detected if joint-calling is performed on a different sample set.

### Estimating allele length

STRling collects additional read evidence for each identified STR locus: reads that completely span the STR locus ("spanning reads") and pairs of reads representing a fragment that spans the locus ("spanning pairs," Fig. 1). Spanning reads are used to determine if an allele shorter than the read length is present at that locus, the size of which can be estimated by taking the average indel size over the locus in all spanning reads.

To estimate the size of alleles greater than the read length, STRling counts the number of STR k-mers in all anchored and overlapping reads assigned to that locus. As an experiment, we edited the reference genome at the Huntington's Disease (HTT) locus to add 300 alleles of varying length between 0 bp (reference allele) and 1800 bp insertion then simulated paired end reads using an empirical insert-size distribution. We found that the simulated allele size was well predicted by the number of anchored reads, or the sum of the k-mers in those anchored reads, up to the median fragment length (Additional file 1: Fig. S1). Beyond the median fragment length, the number of unplaced pairs predicted the allele size. We fitted a linear model to the simulated data and then applied the relationship to new samples to estimate the allele size.

### Outlier detection

STR counts for each locus are normalized by the local sequencing depth to account for differences in library sizes between samples and local sequencing variations.

```
log₂( (sum_str_counts + 1) / local_depth)
```

At each locus STRling tests if the normalized $\log_2$ counts for that sample is greater than the median normalized $\log_2$ counts for all samples. STRling generates z-scores using the median and standard deviation of the normalized counts and corresponding one-sided *p*-values, similarly to previously described outlier scores [22]. These are adjusted for multiple testing across all loci in a given sample using the Benjamini-Hochberg method [50]. A locus is considered significant if the adjusted *p*-value is $< 0.05$.

### Validation

#### *Detecting expansions in novel and reference STR disease loci*

We ran STRling on 134 Illumina PCR-free whole genomes of individuals with known STR disease loci including some unaffected carriers. This cohort had expansions in 14 known STR disease loci, including 83 affected individuals and 11 carriers (Table 1). For FXS there were an additional 17 individuals with premutations, and 22 unaffected family members with alleles in the normal size range. Most of these individuals have been previously described [12, 20–22]. In addition, we included five individuals with CANVAS, one with SCA8, and one with DBQD2.

An STR disease locus was considered found if STRling reported a locus with the disease-causing repeat unit, within 500 bp either side of the position reported in the literature. The set of known STR disease loci that we interrogated can be found in the STRling repository at https://github.com/quinlan-lab/STRling/blob/master/data/hg38.STR_disease_loci.bed. The position error is the distance in base pairs of the STRling call from the position found in the literature. If the STRling call was within the true locus it was given a value of zero. The upstream and downstream sides of the locus were compared separately, then the maximum of these values was taken to be the position error for that locus. The same method was used to evaluate ExpansionHunter Denovo calls.

#### *Comparison to other novel STR detection methods*

We ran ExpansionHunter Denovo v0.9.0 [26] in outlier mode on the same 134 genomes used to validate STRling (see above), referred to as "EHdn affected vs. affected". Additionally, each sample was run in outlier mode with a set of 260 unaffected controls, "EHdn affected vs. controls" (full workflow: https://github.com/hdashnow/longSTR/blob/master/EHdn_1vsControls.groovy). The commands used were:

```
ExpansionHunterDenovo profile --reads $input.cram --refer-
ence Homo_sapiens_assembly38.fasta --output-prefix $sample
ExpansionHunterDenovo  merge  --reference  Homo_sapiens_
assembly38.fasta --manifest $input.manifest --output-pre-
fix all
python outlier.py locus --manifest $input.manifest --mul-
tisample-profile $input.json --output $output.tsv
```

An STR disease locus was considered found if ExpansionHunter Denovo reported a locus with the disease-causing repeat unit, within 500 bp either side of the position reported in the literature.

*Comparison to long reads: Ashkenazim trio*

Illumina reads from the 300X Ashkenazim trio were downloaded from Genome In a Bottle (https://github.com/genome-in-a-bottle/giab_data_indexes), aligned with BWA MEM, and subsampled to approximately 30X using `samtools view -h -C -s` for a final mean sequencing depth of HG002 (son) 30.01X, HG003 (father) 33.20X and HG004 (mother) 33.96X.

We obtained assemblies of HiFi reads from the same Ashkenazim trio from PacBio. Individuals were assembled using PacBio Improved Phased Assembler with default settings (see Additional file 1: Supplementary Methods). Contigs were aligned to GRCh38 with pbmm2 then variants were called with bcftools mpileup. We then limited the callset to insertions greater than 10 bp and counted the most frequent 1–6 bp k-mer in each. Each STRling call from above was annotated with the closest PacBio insertion. We additionally counted the number of times the STRling repeat unit was found in the PacBio insertion. Before calculating true and false positives, we removed STRling calls overlapping segmental duplications, LCRs, centromeres, and telomeres and limited the calls to those on chromosomes chr1-22, X, and Y (excluding others, such as alt and decoy contigs). A STRling call was considered a true positive if it had a pacbio insertion with a matching most frequent k-mer, or if at least 50% of the PacBio insertion was made up of the STRling repeat unit. All other STRling calls with at least one overlapping PacBio contig but no matching variant call were considered false positives. Code for PacBio alignment, variant calling, k-mer counting, and comparison to STRling calls can be found at https://github.com/hdashnow/longSTR.

STRling outlier results for the Ashkenazim trio were classified as Mendelian matches if the child's alleles matched inheritance expectations, or Mendelian violations otherwise (code: https://github.com/laurelhiatt/strling-MV). STRling alleles were considered matched if their sizes were within 25% of the parent allele or ten bp. Only loci with at least depth of 15 reads and no missing alleles were considered. STRling calls were compared to variants called from PacBio assemblies of the same individuals. PacBio variants were considered Mendelian matches if both child alleles of the designated repeat unit were within 10bp or 25% in size to matched parent alleles, and a Mendelian violation if this was not the case. The % difference in allele size between each parent-child pair was calculated as such: (parent - child)/parent.

## Supplementary Information

---

Additional file 1: Table S1. Novel STR diseases. Table S2. Sensitivity of ExpansionHunter denovo run on PCR-free Illumina WGS of 94 subjects with alleles of pathogenic size at an STR disease locus. EHdn was run in outlier mode once for each subject against 260 individuals from the 1000 genomes project. Table S3. STRling false discovery rate when compared to long reads. STRs were called using STRling on ~30x Illumina WGS of the Ashkenazim trio. STRling calls >20 bp were verified by if there was a matching insertion in the PacBio HiFi assembly from the same sample (see Methods). Figure S1. Simulated allele size at the HTT locus predicts the number of anchored and unplaced pairs. Figure S2. STRling sensitivity (A) and false discovery rate (B) as a function of allele size. Figure S3. STRling underestimates allele sizes in Fragile X Syndrome (FXS). Figure S4. STRling locus discovery and outlier detection by sample size. Figure S5. STRling resource usage. Figure S6. ExpansionHunter Denovo resource usage. Supplementary Methods.

Additional file 2. Review history.

## Acknowledgements
The authors would like to thank Zev Kronenberg of PacBio for assembling the HiFi Ashkenazim trio and providing advice on long-read analysis. Michael A. Eberle and Egor Dolzhenko of Illumina Inc. kindly provided access to the genomes from their ExpansionHunter and ExpansionHunter Denovo papers.

## Review history
The review history is available as Additional file 2.

## Peer review information
Andrew Cosgrove and Stephanie McClelland were the primary editor of this article and managed its editorial process and peer review in collaboration with the rest of the editorial team.

## Authors' contributions
HD, BSP, JB, and ARQ developed STRling. HD and ARQ wrote the manuscript. HD, LH, and JB performed computational analyses. SJB, GR, MD, and NGL were involved in the sequencing and characterization of positive control participants. PL recruited the SCA8 participant. RHR and RJR recruited CANVAS participants. AJL and HCM sequenced and characterized the DBQD2 participant. All authors read and approved the manuscript.

## Funding
Support for this work was provided by the National Institutes of Health, National Heart, Lung, and Blood Institute, through BioData Catalyst program award 1OT3HL142479-01, 1OT3HL142478-01, 1OT3HL142481-01, 1OT3HL142480-01, 1OT3HL147154 to HD and NIH awards R01HG010757 and R01GM124355 to ARQ. GR is supported by the Australian NHMRC (Investigator Grant, APP2007769). NGL is supported by NHMRC Principal Research Fellowship APP1117510. This work is also supported by the MRFF Genomics Health Futures Mission Grant 2007681 (GR and NGL). Any opinions expressed in this document are those of the authors and do not necessarily reflect the views of NHLBI, individual BioData Catalyst Consortium members, or affiliated organizations and institutions. The computational resources used were partially funded by the NIH Shared Instrumentation Grant 1S10OD021644-01A1.

## Availability of data and materials
STRling is open source and freely available at: https://github.com/quinlan-lab/STRling under MIT license [30]. The specific version of STRling used in these analyses is available at: https://zenodo.org/record/6819612 [51].
PCR-free WGS from participants with CANVAS and SCA8 are available from the Sequence Read Archive (SRA, https://www.ncbi.nlm.nih.gov/sra) project PRJNA885420, SRA accessions SRR21753323-SRR21753328 [52]. PCR-free WGS from the participant with Baratela-Scott is available from the database of Genotypes and Phenotypes (dbGAP, https://www.ncbi.nlm.nih.gov/gap/) project phs000693, sample UW04-1 [53].
The following genomes analyzed in this study have been published previously:
PCR-free WGS of the ten test samples is available from the Sequence Read Archive, accession SRP148723 (individual sample accessions SRX4114164-SRX4114173) [22, 54].
PCR-free HiSeqX whole genome sequence data on 1 sample with triplet repeat expansions (premutation and full expansions) Supplemental_Table_7 in the ExpansionHunter paper [20]. WGS available at: https://ega-archive.org/datasets/EGAD00001003562 [55].
Ashkenazim trio: PacBio HiFi reads on SRA [35]. HG002: SRR10382244, SRR10382245, SRR10382248 and SRR10382249 [56]. HG003: SRR11567494 - SRR11568082 [57]. HG004: SRR11568075 - SRR11568077, and SRR11568083 - SRR11568088 [58]. Illumina reads are available from: https://github.com/genome-in-a-bottle/giab_data_indexes [59].

# Declarations

## Ethics approval and consent to participate
This study was approved by the University of Western Australia Human Research Ethics Committee (IRB approval number RA/4/20/1008) and the University of Washington (IRB approval number 28853). All individuals have given written informed consent for participation and publication. The experimental methods in this study comply with the Helsinki Declaration.

## Competing interests
The authors declare that they have no competing interests.

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

## 
