## [Additional file 2. Review history. · Genome Biology]

Review History

First round of review

Reviewer 1

Were you able to assess all statistics in the manuscript, including the appropriateness of statistical tests used? Yes: No additional statistical review is needed.

Were you able to directly test the methods? Yes

Comments to author:

Dashnow et al developed a new tool for detecting short tandem repeat expansions based on k-mer counting. The authors showed that their method, STRling, is able to detect both known and novel loci with expanded repeat size (defined by an outlier detection method). The sensitivity for detecting expanded repeats in known STR loci is reasonably high, except particularly underperformed on repeat loci with high GC content. Other than being able to locate expanded repeats, another strength of this method is the ability to joint-call multiple samples, which helps further refining the mapped location. This new method provides substantial improvement over comparable existing tools. The manuscript is well written and easy to follow. However, there are several things that the authors should clarify and consider further improvement.

Specific comments:

1. In theory, STRling should be able to detect any repeats of any size, but it is unclear what is the size distribution of repeats that are detected from a whole genome. It should be able to address from the Ashkenazim trio analysis, but the authors chose to analyze calls that are with minimum size of 50bp on the autosomes. How is 50bp determined? What if repeats of <50bp are included? Why only restrict to autosome when it can detect repeat expansion diseases on X chromosome (e.g. FXS and SBMA)?
2. Similarly, why 10bp insertion size was selected from pacbio variants as positive validation? Can't they compare the size estimation between the STRling calls and pacbio variants as they did in Figure 4?
3. It is expected that smaller (e.g. within the read length) repeats are easier to pick up by short-read sequencing. Please provide more details on the accuracy in different repeat size ranges, e.g. <50bp, 50-150bp, 150-450bp and >450bp. Please also compare with the corresponding insertion size detected by pacbio.
4. The consideration of Mendelian concordance and violation seems to be too simplistic. As mentioned above, using 10bp insertion size is arbitrary. This selected size may work for large repeats, but probably too stringent for small repeats. It is hard to evaluate without knowing the size distribution of repeats detected and analyzed.
5. What is the cause of invalidation of de novo calls? Is it the false positive in the child or the false negative in the parents? Again, what is the size range of all these de novo and inherited variants?

6. Why did they downsample the trios read depth to 70x? Shouldn't the typical average read depth for whole genome sequencing be ~30x? How does the read depth impact the number or accuracy of the calls? Does joint-calling make any difference on total number of repeats detected per genome?

7. What is the cause of high error position estimation for DM2 specifically (Figure 3)? Why can't it be improved even with joint-calling? Please discuss.

8. When they compare the mapped position accuracy, did they compare bounds on both ends or just one end of the bounds?

9. In theory, their method should be able to size both two alleles in a repeat tract (e.g. the presence of spanning reads). However, based on the inability to differentiate carrier status for recessive diseases (Table 2), the authors probably didn't do it. Can they explain why?

10. The authors stated that the SCA6 expansion is difficult to detect due to the expanded allele only being 26bp greater than the reference allele. What is the lower limit of detection in cases like this? For example, HTT patients can have 36 repeats (108bp), while the reference genome has 21 repeats (63bp). Will the 45bp difference between non-pathogenic and pathogenic length be detectable?

11. The authors stated that "134 subjects tested had a median of 9 (1-252) significant STR expansions each." - How does STRling handle cases where the same motif is expanded at more than one locus in the genome? This might be rare in mendelian disorders but could happen in cases like cancers.

12. It would be good to know how much of the sink regions identified by STRling are actually known repeat regions. If most of them are already known, then it seems that the tool is more specific to the known loci than the novel ones. One way to check this is by removing some of the sink regions that are already known, run the tool and see if it could pick the repeats up.

13. While they reported false discovery rate on the 240 samples from 1000 Genomes Project, they have not mentioned if there were outlier repeats detected other than the known expanded repeats in the 94 disease samples.

14. Since the tool can summarize all k-mers detected in the repeat tract, it would be helpful to provide proportion of the major motif and composition of other minor repeats. This may allow detection of interrupting sequence, which is something missing currently in their analysis.

Reviewer 2

Were you able to assess all statistics in the manuscript, including the appropriateness of statistical tests used? Yes: no concern

Were you able to directly test the methods? No

Comments to author:

This manuscript describes a k-mer counting approach to detect short tandem repeat expansions (STRs). One unique feature of the method STRling is that it can call expansions at both known and novel STR loci. The main idea is that STRling uses k-mer counting to find

all the reads with substantial STR content, and after these candidate reads are collected, it then uses their well-mapped "mates" to assign them to their correct locus.

Background section, they say "48 STR expansions cause Mendelian human diseases". However, there is no reason why STR does not cause common and complex human diseases. In fact, quite a few STRs are already associated (increase the risk, but less than 100% penetrance) with ALS and bipolar disorder and other diseases.

Overall the background section is written in a way that is not suitable for typical Genome Biology readers. It mainly talks about a few computational methods for STR allele detection (even so the list is not comprehensive), without much details on the biological relevance of STR expansions. There are also issues in educating readers that the reference genome is built from healthy individuals with small number of repeats in STR, or with no annotated repeats in specific regions that can be expanded in diseases such as CANVAS or FAME. There is no background description on the difference on typical STRs (such as polyQ) and more complex STRs (such as FAME mixed-motif repeats or Fragile X CGG ones) that may result in completely different computational strategies or completely different experimental protocols. It also almost completely ignored the latest technical platforms such as PacBio HiFi (when accuracy is as good as or better than Illumina) or Bionano or even the recently announced Illumina Infinity sequencing that can probably readily address many of the problems that this manuscript is trying to address. In summary, it is just not informative enough for

While the authors emphasized on the importance of mapping errors on STR detection, it is strange that the input to STRling are reads that are aligned to known STR regions, or are unmapped reads, from a BAM or CRAM file. Most repeat detection algorithms (or even some SV detection algorithms) suggest a different set of parameters for alignment, to improve the quality of STR discovery and quantification. The impact of alignment software (and specific sets of parameters) need to be taken into account in the analysis.

The authors used a set of filters to drop down candidate regions significantly. On average, the 134 subjects tested had a median of 9 significant STR expansions each. It is difficult to judge the reliability of such calls without experimental validation, but one easy thing to do is to assess similar number of subjects in the same sequencing batch who are not carriers or not affected, and see how the results differ.

The claim about "PCR is the current standard method for STR disease diagnostics" is not correct. It may be the case for certain loci in certain diseases (such as many polyQ diseases where the expanded repeats are merely a few hundred base pairs), but definitely cannot be extrapolated to all repeat expansion disorders. Some can benefit from repeat primed PCR (which is different from PCR), with pulse field, yet others depend on Southern blot or capillary electrophoresis, and some others depend on highly specialized enzymes for PCR, and some depends on other more modern method such as molecular combing. It is all just very much disease dependent and loci dependent. The diseases assays in this manuscript (for example, Figure 3) only represent a portion of the diseases where repeat plays a role. This is something not made explicit and give readers a false impression that NGS can actually solve the problem of repeat expansions, which is not the case; considering that so many similar software tools have already been published yet the true clinical problem has not actually been solved yet, it highlight the challenge in even using NGS to study these diseases or to find novel STRs. Having said that, I also acknowledge that there are specific application scenarios

(one example is forensic diagnosis) where the labs began to design specialized assays and then use NGS to do STR typing, but then again it is not the same as whole-genome NGS.

I had difficulty with the entire section of "Long reads enable estimates of STRling's false discovery rate". I would have expected (when reading earlier portion of the manuscript) that if PacBio HiFi is used, the authors would be able to either identify known STRs from patients, or find novel (not present in the reference genome) STRs that may lead to human diseases. Yet the entire section is about "false discovery rate": not that it is not important, but that this is not one typical reader expect. Many of disease-causal repeat expansions that are discovered in the past 3 years (for example, CANVAS and FAME if I remember correctly though I could be wrong) are actually found by long reads, not short reads, and they would have served as perfect examples of how STRling can possibly work on long read. Yet the whole section is about healthy individuals such as Ashkenazim trio and I do not really see the real point of this section here compared to the real clinical problem that a method such as STRling is trying to solve.

There is a general lack of comparison to existing computational tools. This field is flooded with bioinformatics methods and it would be ideal to do more thorough comparison to show what are the advantages. There are some comparisons such as ExpansionHunter Denovo but certainly not enough, and there are sections such as "algorithm resource requirements" where there is basically no comparison so a reader would have no real idea whether the time/memory requirement improves over existing tools or is excessive. In fact, I would argue that requirement about the joint merge stage (29.04 GB) and the joint outlier stage (27.05 GB) sound quite resource-intensive and not really applicable to typical labs unless dedicated servers are used.

There are many places that can be considered as over statement. Take an example in Abstract: "It is the first method to resolve the position of novel STR expansions to base pair accuracy", it can certainly be challenged by other authors whether short or long read or pseudo-long read were used in previous publications, there are many previous examples of finding novel STR expansions with base pair resolution which is exactly why so many repeat expansion diseases are actually discovered over the past few years.

Responses to reviewer comments

Reviewer #1: *Dashnow et al developed a new tool for detecting short tandem repeat expansions based on k-mer counting. The authors showed that their method, STRling, is able to detect both known and novel loci with expanded repeat size (defined by an outlier detection method). The sensitivity for detecting expanded repeats in known STR loci is reasonably high, except particularly underperformed on repeat loci with high GC content. Other than being able to locate expanded repeats, another strength of this method is the ability to joint-call multiple samples, which helps further refining the mapped location. This new method provides substantial improvement over comparable existing tools. The manuscript is well written and easy to follow. However, there are several things that the authors should clarify and consider further improvement.*

Specific comments:

1. In theory, STRling should be able to detect any repeats of any size, but it is unclear what is the size distribution of repeats that are detected from a whole genome. It should be able to address from the Ashkenazim trio analysis, but the authors chose to analyze calls that are with minimum size of 50bp on the autosomes. How is 50bp determined? What if repeats of <50bp are included? Why only restrict to autosome when it can detect repeat expansion diseases on X chromosome (e.g. FXS and SBMA)?

Thank you for asking these helpful questions. We have included the X and Y chromosome variants in our updated analysis. Supplementary Table 3 has been updated accordingly. The two tables below present the refined analysis using 30X sequencing coverage, with and without sex chromosomes so that you can compare them to each other and the original 70X analysis without sex chromosomes. Looking at the 2-6 bp loci, the addition of sex chromosomes added 6 false positives and 6 true positives, resulting in a slight increase in the false positive rate from 0.15 to 0.17 (0.145 to 0.16 for significant outliers), driven by false positives on the Y chromosome. Note that variant detection on the Y chromosome is notoriously difficult.

The 50 bp minimum size is based on previous experience with STR variant calling. This is the threshold under which within-read STR methods (HipSTR, LobSTR, etc.) perform well. >50 bp also includes the majority of pathogenic STR alleles (other than SCA6). When designing STRling we were targeting this >50bp variant size by focusing on signals such as highly repetitive reads and anchored pairs. This signal requires larger allele sizes. We feel that this >50bp size range is a more fair test of STRling's performance in its intended context.

FDR aggregated over all samples: 30X, chr 1-22, X and Y				
	all loci	2-6bp	significant outliers	significant outliers 2-6bp
FDR	0.4652129	0.1683168	0.4799618	0.1617647
TP	1030	252	545	171
FP	896	51	503	33

FDR aggregated over all samples: 30x, autosomes (chr 1-22 only)				
	all loci	2-6bp	significant outliers	significant outliers 2-6bp
FDR	0.4636364	0.1546392	0.479723	0.1450777
TP	1003	246	526	165
FP	867	45	485	28

2. Similarly, why 10bp insertion size was selected from pacbio variants as positive validation? Can't they compare the size estimation between the STRling calls and pacbio variants as they did in Figure 4?

When comparing STRling calls on short reads with indels called in long reads we matched the variants by looking for the most common sequence motif. For STRling calls this is the repeat unit. For PacBio calls this is done by counting the number of times each k-mer is present in that insertion. We did not feel we could do this reliably in very short sequences. Given all the STRling calls were greater than 20 bp, 10 bp seemed a reasonable compromise between sufficient sequence to operate on and likely to capture most loci that were also called by STRling. It is also consistent with the value used in our comparisons of *de novo* mutations; see below.

3. It is expected that smaller (e.g. within the read length) repeats are easier to pick up by short-read sequencing. Please provide more details on the accuracy in different repeat size ranges, e.g. <50bp, 50-150bp, 150-450bp and >450bp. Please also compare with the corresponding insertion size detected by pacbio.

Good question. In summary, we found no difference in STR STRling's FDR with respect to size estimated from PacBio.

STRling is specifically designed to complement the functionality of within-read STR callers by detecting large STR expansions which will most commonly exceed the read lengths achieved by Illumina sequencing. Larger alleles would be expected to have more read evidence. Therefore we would not expect short alleles to be easier to detect with our method. However, to measure this, we took the set of STRling 2-6bp repeat unit variants assessed for FDR in Table S3 and divided them into 10 equally sized groups (deciles) based on STRling allele size. We calculated the FDR in each decile by comparing STRling calls to

PacBio calls. As demonstrated in the following figure (now added as Supplementary Figure 2B), we did not observe a relationship between allele size and FDR.

The set of STRling 2-6bp repeat unit variants assessed for False Discovery Rate (FDR) using PacBio (Supplementary Table 3) were divided into ten equally sized groups (deciles) based on STRling predicted allele size. We report the FDR for each decile.

4. The consideration of Mendelian concordance and violation seems to be too simplistic. As mentioned above, using 10bp insertion size is arbitrary. This selected size may work for large repeats, but probably too stringent for small repeats. It is hard to evaluate without knowing the size distribution of repeats detected and analyzed.

Thank you for catching this. There was a detail missing from the methods describing how we determine concordance. Alleles were considered concordant if they were within 10 bp or 25%. For example, a 100 and 125 bp variant would be treated as concordant, while a 100 and 150 bp variant would be flagged as a mendelian violation. This has been added to the methods.

The 10bp/25% thresholds were chosen by plotting the absolute and % difference in allele sizes between the child and the most similar parent allele at all loci (Figure below). We chose thresholds that would assign the majority of the data to mendelian concordance, on the assumption that true de novo expansions would be rare. I have additionally provided the distribution of allele sizes for reference to aid in evaluation of this point (Figure below).

For each parent-child pair in a set of 22 trios we compared the largest allele predicted by STRling at each locus and calculated the A: absolute difference, $\text{abs}(\text{parent}-\text{child})$ and the B: % difference, $(\text{parent} - \text{kid})/\text{parent} * 100$. Each line represents a parent-child pair from a set of 22 unpublished PCR-free WGS trios (these individuals are not part of the present paper). We found that the majority of differences were near 0 bp or 100%, suggesting general concordance. Using a threshold of 10 bp or 125% includes the majority of the calls.

Allele size distribution of STRling calls in the HG002, HG003, HG004 trio used for FDR estimation with PacBio sequencing.

5. What is the cause of invalidation of de novo calls? Is it the false positive in the child or the false negative in the parents? Again, what is the size range of all these de novo and inherited variants?

Thank you. To assess this, we manually examined the PacBio assembled contigs for each predicted de novo expansion in IGV and noted the reason for invalidation of a de novo predicted by STRling. For several loci there were numerous mismatches between the PacBio contig and the reference genome in the region surrounding the STR. Upon closer

inspection, these also overlapped annotated segmental duplications. We then excluded segmental duplications from our analyses (and updated the Results accordingly). For the remaining sites, the main cause of invalidation of STRling de novo calls was the presence of an expansion in one or both of the parents that were found by PacBio but missed by STRling. In our original analysis, we required at least two contigs in each parent before considering the site for validation. We noticed that in many cases the parents had more than two contigs at a single site, indicating either paralogy or that the PacBio assembly produced more than two alleles at that position, thereby confusing the interpretation of allele status. We have repeated this analysis with more stringent requirements and reported the updated numbers in the Results section.

The size range of the variants considered was: 22 to 719 bp. Of those, the mendelian matches covered the full range of alleles sizes while the mendelian violations ranged from 25 to 303 bp.

6. Why did they downsample the trios read depth to 70x? Shouldn't the typical average read depth for whole genome sequencing be ~30x? How does the read depth impact the number or accuracy of the calls? Does joint-calling make any difference on total number of repeats detected per genome?

Great point. We have repeated this analysis with the reads downsampled to ~30X. This decreased the total number of STRling calls from a mean of 616 to 297 per individual (for autosomes only, the final numbers reported in the paper include chr X and Y). However, most of these were homopolymers. The number of 2-6 bp repeat unit STRling calls changed from a mean of 77 to 62. Of these 47 and 42 respectively were significant. **Reducing the sequencing depth from ~70x to ~30x reduced the overall FDR from 0.48 to 0.41 and the FDR for 2-6 bp repeat unit loci from 0.20 to 0.16.** Across the entire trio, the total number of true positive significant 2-6 bp repeat unit loci reduced from 117 to 108. The comparisons here still exclude the X and Y chromosomes. The final numbers reported in the paper include the X and Y, and so vary slightly from those reported here.

Joint calling typically increases the number of repeat loci reported per genome, because if an expansion is found in any one individual, that site is interrogated in all samples (see Supplementary Figure 5, panel E). To more thoroughly answer this question we have performed additional joint calling analyses and added them to the paper. We additionally explored the effect of cohort size on outlier testing. We performed the full joint-calling workflow on 20, 100, 200, 500, and 1000 genomes. Increasing cohort size increased the number of significant outliers reported per individual. After filtering to 2-6 bp repeat units outside highly repetitive regions (LCRs, segmental duplications, centromeres, and telomeres) this amounted to an average of 10.2 significant outliers in a cohort of 20 compared with 33.6 in a cohort of 1000 (Supplementary Figure 4).

We randomly sampled individuals from the 1000 Genomes Project, performed STRling joint calling on each subset, and reported the number of significant outliers per individual. All outliers on canonical chromosomes chr1-22, X, and Y, outliers at 2-6bp repeat unit loci, and outliers at 2-6bp repeat unit loci excluding those overlapping LCRs, segmental duplications, centromeres, or telomeres. The inset shows the same data but with differing y axes.

7. What is the cause of high error position estimation for DM2 specifically (Figure 3)? Why can't it be improved even with joint-calling? Please discuss.

Good question. DM2 locus is a compound/complex repeat, with the form $(TG)_n(TCTG)_n(CCTG)_n$ in the direction of transcription, and $(CAGG)_n(ACAG)_n(AC)_n$ in genomic coordinate order. Expansions in the CCTG/CAGG tract are associated with DM2. However, the other repeat units are also variable.

STRling called expansions in both CAGG and ACAG in the affected individual at the DM2 locus. However, as you can see in IGV below, there was soft-clipped read evidence for the ACAG repeat but not the CAGG repeat. Because STRling treated each locus independently, it only used anchored read evidence to approximate the position of the CAGG expansion, resulting in a less accurate position. There is only one individual in our cohort with a pathogenic DM2 expansion. The position accuracy was not improved by adding additional unaffected individuals.

We have added an additional discussion of this locus in the paper:

The greatest joint-calling position error was observed at the DM2 locus, which is a complex locus with the form $(TG)_n(TCTG)_n(CCTG)_n$, with CCTG expansions associated with disease. We had only one individual in our cohort with DM2, and no soft-clipped read evidence was observed for the CCTG expansion. Therefore, joint calling was not able to improve the position estimate.

8. When they compare the mapped position accuracy, did they compare bounds on both ends or just one end of the bounds?

This is an important detail that we should have emphasized more in the paper. Position accuracy was computed by comparing both ends and reporting the largest discrepancy.

From the methods:

“An STR disease locus was considered found if STRling reported a locus with the disease-causing repeat unit, within 500 bp either side of the position reported in the literature. The set of known STR disease loci that we interrogated can be found in the STRling repository at https://github.com/quinlan-lab/STRling/blob/master/data/hg38.STR_disease_loci.bed. The position error is the distance in base pairs of the STRling call from the position found in the literature. If the STRling call was within the true locus it was given a value of zero. The left and right sides of the locus were compared separately, then the maximum of these values was taken to be the position error for that locus. The same method was used to evaluate ExpansionHunter Denovo calls.”

I have added a summary of this comparison metric in the Figure 2 legend.

“Max position error is the position difference between the known and predicted locus (max of left and right). Zero indicates the predicted position is within or at the bounds of the known locus.”

9. In theory, their method should be able to size both two alleles in a repeat tract (e.g. the presence of spanning reads). However, based on the inability to differentiate carrier status for recessive diseases (Table 2), the authors probably didn't do it. Can they explain why?

STRling quantifies and reports evidence for both alleles exceeding the read length and alleles within the read length, including reporting the number of spanning reads and spanning pairs. However, when an individual has two alleles that exceed the read length, it cannot distinguish the sizes of the two large alleles. This is because our allele size estimates are made by summing the read evidence at the locus; we have found that there simply isn't enough confident signal to distinguish the signal from two distinct, expanded haplotypes. Anchored pairs cannot be assigned to a specific allele if both are large (Figure 1C).

While it is difficult to clearly call zygosity using STRling, we made an educated guess by inferring that loci with no spanning reads and no or few spanning pairs are likely to be homozygous or hemizygous for a large expansion. When applying this approach to the true positive probands, 55/69 STRling calls had the expected zygosity for the disease mode of inheritance (heterozygous for dominant conditions, homozygous for recessive, hemizygous for X-linked in males). Only probands were checked for zygosity because while we know which other family members tested positive for an expansion by PCR, we do not know their expected zygosity due to missing phenotype and pedigree information.

Abbreviated text included in Results:

While STRling does not directly report zygosity, we made an educated guess by inferring that loci with no spanning reads and no or few spanning pairs are likely to be homozygous or hemizygous for a large expansion. When applying this approach to the true positive probands, 55/69 STRling calls had the expected zygosity for the disease mode of inheritance (heterozygous for dominant conditions, homozygous for recessive, hemizygous for X-linked in males).

Using the same approach we predicted zygosity using STRling in the 260 individuals from the 1000 Genomes that had significant outliers in a pathogenic locus. Almost all STRling calls were predicted to be heterozygous, with the exception of one FTDALS1 call and all the DBQD2 calls, which had no spanning reads and so appeared homozygous. For DBQD2 this is expected because the expansion is contained within a 238 bp non-repetitive novel inserted sequence and normal individuals have an AAAAG STR within that insertion. Given the size of the locus in even unaffected individuals, we would be surprised to find spanning reads in any individual, regardless of affected status.

This result has been added to the Results section, including a column added to Table 2, also included below.

Table 2: Significant outliers called by STRling in 260 individuals from the 1000 Genomes Project.

We performed joint-calling and outlier testing, then filtered the results to 32 well-characterized known pathogenic STR loci (see **Methods**). There were 204 significant STRling calls across ten loci (the others had no significant calls), resulting in an estimated FDR of 0.078. Of these, most also had evidence of a short, likely non-pathogenic allele (reads and read pairs spanning the locus), suggesting heterozygosity. AD: Autosomal Dominant, AR: Autosomal Recessive, XD: X-linked Dominant, XR: X-linked Recessive. Novel STR disease loci (not in the reference genome) are indicated in bold/underline. *No spanning reads, spanning pairs only.

Disease	Significant outlier	Evidence of a short allele	Proportion significant outlier	STRling est. > pathogenic threshold	Inheritance	Notes
CANVAS	12	12	0.0462	0	AR	0.7% carrier frequency [9]
DBQD2	5	5*	0.0192	0	AR	
DM2	7	7	0.0269	0	AD	Age onset: ~30-40
FRA12A	57	57	0.219	0	AD	
FRAXE	7	7	0.0269	0	XR	
FRDA	61	61	0.235	1	AR	Age onset: 5-25
FTDALS 1	33	32 (1*)	0.127	0	AD	Age onset: 27-85
FXS	0	N/A	0	0	XD	Multiple syndromes with varying pathogenic size thresholds and age of onset: FXS 2, FXTAS 60-65, POI?
SCA10	8	8	0.0308	0	AD	Age onset: 12-48
SCA8	14	14	0.0538	1	AD	Age onset: 1-73

10. The authors stated that the SCA6 expansion is difficult to detect due to the expanded allele only being 26bp greater than the reference allele. What is the lower limit of detection in cases like this? For example, HTT patients can have 36 repeats (108bp), while the reference genome has 21 repeats (63bp). Will the 45bp difference between non-pathogenic and pathogenic length be detectable?

We plotted all the pathogenic expansions by total allele size in bp reported by PCR and categorized them by whether STRling detected the expansion from individual calling. STRling missed alleles smaller than 130bp, and was most likely to detect alleles larger than 150bp. Most of the calls in this size range are HD/HTT. TRF reports 64bp of CAG in hg38 at this position, so the 130-150bp calls from PCR would represent 66-86bp insertions. In summary, STRling appears to be most sensitive to insertions greater than ~90bp and may struggle to detect smaller pathogenic variants given a single genome. However, STRling was

able to correctly identify all pathogenic HD expansions in this cohort (the smallest being 132bp/69bp insertion) once joint calling and outlier testing was applied (Table 1).

Orthogonally validated true positive pathogenic STR expansions plotted by total allele size in bp reported by PCR and categorized by whether STRling detected the expansion with individual calling. STRling missed alleles smaller than 130bp, and was most likely to detect alleles larger than 150bp. Most of the calls in this size range are HD.

11. The authors stated that "134 subjects tested had a median of 9 (1-252) significant STR expansions each." - How does STRling handle cases where the same motif is expanded at more than one locus in the genome? This might be rare in mendelian disorders but could happen in cases like cancers.

We agree this case should be explained in more detail. STRling uses the position of uniquely aligned read pairs to correctly identify the region of the genome that a specific repetitive read arose from. In the majority of cases, there is sufficient unique sequence to identify the source locus. For very large STR expansions (i.e., those that are larger than the sequencing library insert size), there will be a number of pairs for which both reads are completely repetitive. In these cases, we may not be able to identify the source position of the read pair unless the particular repeat expansions were unique. In such cases of multiple large expansions with the same repeat unit, some reads would not be assigned; consequently, we would underestimate the size of the allele. However, since many anchored pairs would be present at the edges of the event, we would still correctly identify the position of each expansion.

We have added the following to the Results section:

If two large expansions (greater than the insert size) with the same repeat unit were present with the same repeat unit there will be a number of pairs for which both reads are completely repetitive. In these cases, we may not be able to identify the source position of the read pair. Some reads would not be assigned; consequently, we would underestimate the size of the allele. However, since many anchored pairs would be present at the edges of the event, we would still correctly identify the position of each expansion.

12. It would be good to know how much of the sink regions identified by STRling are actually known repeat regions. If most of them are already known, then it seems that the tool is more specific to the known loci than the novel ones. One way to check this is by removing some of the sink regions that are already known, run the tool and see if it could pick the repeats up.

Good question. The STR sink regions identified by STRling are a subset of known and annotated STR loci that are both large and predominately perfect repeats. We consider any reads aligned to these highly repetitive regions to have suspect mapping positions, and so flag them for k-mer counting and potential alignment correction based on read pair information. These regions in and of themselves are not necessarily interesting or loci of variation (although they could be), rather they are an algorithmic shortcut to boost the recovery of informative sequence reads for expansion discovery.

We have edited the initial discussion of sink loci in the paper to clarify:

“Reads containing substantial STR content will tend to map to the position in the reference genome with the longest matching repeat; we define such loci as STR “sinks”. These long perfect repeat tracts act as sinks to which STR-containing reads disproportionately map. However, because STRs occur throughout the genome, the longest locus is unlikely to be the one from which that read truly originated.”

13. While they reported false discovery rate on the 240 samples from 1000 Genomes Project, they have not mentioned if there were outlier repeats detected other than the known expanded repeats in the 94 disease samples.

There was an average of 1 significant expansion in known pathogenic loci per individual in this cohort and on average 9 significant outliers per individual including pathogenic and other STR loci using recommended filters (excluded homopolymers, seg dups, LCRs, centromeres, and telomeres). While most individuals had 1 significant expansion at a pathogenic locus, some had none, and few had 2, 3, or 4 calls.

14. Since the tool can summarize all k-mers detected in the repeat tract, it would be helpful to provide proportion of the major motif and composition of other minor repeats. This may allow detection of interrupting sequence, which is something missing currently in their analysis.

This would be an excellent addition to future iterations of the STRling algorithm. As written STRling only retains the most common k-mer, therefore this would be a major rewrite of the core algorithm. We will add this to a list of future developments.

Reviewer #2: *This manuscript describes a k-mer counting approach to detect short tandem repeat expansions (STRs). One unique feature of the method STRling is that it can call expansions at both known and novel STR loci. The main idea is that STRling uses k-mer counting to find all the reads with substantial STR content, and after these candidate reads are collected, it then uses their well-mapped "mates" to assign them to their correct locus.*

1. Background section, they say "48 STR expansions cause Mendelian human diseases". However, there is no reason why STR does not cause common and complex human diseases. In fact, quite a few STRs are already associated (increase the risk, but less than 100% penetrance) with ALS and bipolar disorder and other diseases.

Thank you for this suggestion. We have added the following to the introduction:

“STRs have been associated with autism, intelligence and depression [3–6]. Supporting a mechanistic link, STR variation has been associated with expression levels of nearby genes [7].”

2. Overall the background section is written in a way that is not suitable for typical Genome Biology readers. It mainly talks about a few computational methods for STR allele detection (even so the list is not comprehensive), without much details on the biological relevance of STR expansions. There are also issues in educating readers that the reference genome is built from healthy individuals with small number of repeats in STR, or with no annotated repeats in specific regions that can be expanded in diseases such as CANVAS or FAME. There is no background description on the difference on typical STRs (such as polyQ) and more complex STRs (such as FAME mixed-motif repeats or Fragile X CGG ones) that may result in completely different computational strategies or completely different experimental protocols. It also almost completely ignored the latest technical platforms such as PacBio HiFi (when accuracy is as good as or better than Illumina) or Bionano or even the recently announced Illumina Infinity sequencing that can probably readily address many of the problems that this manuscript is trying to address. In summary, it is just not informative enough for

Thank you for these suggestions. We have elaborated upon the following topics in the introduction. For example, adding the following:

Disease mechanisms include polyglutamine aggregation, RNA toxicity, altered methylation, and repeat-associated non-ATG translation.

For example, in CANVAS the non-pathogenic AAAAG STR found in the reference is replaced by an AAGGG repeat, which, when expanded, causes disease. In Baratela-Scott Syndrome the pathogenic expansion occurs within a 238 bp insertion relative to the reference genome. Finding a novel STR may indicate that the reference was generated from an individual with an alternate haplotype, or that an error occurred in the assembly of the reference genome.

While long-read sequencing strategies like PacBio and Oxford Nanopore are increasing in popularity, they are still prohibitively expensive for large-scale genomic studies. Additionally, we need methods to analyze the many short-read genomes of patients with unsolved rare diseases.

In addition, we included a discussion of the complex DM2 locus in the results section as part of a discussion of position accuracy:

The greatest joint-calling position error was observed at the DM2 locus, which is a complex locus with the form $(TG)_n(TCTG)_n(CCTG)_n$, with CCTG expansions associated with disease. We had only one individual in our cohort with DM2, and no soft-clipped read evidence was observed for the CCTG expansion. Therefore, joint calling was not able to improve the position estimate.

3. While the authors emphasized on the importance of mapping errors on STR detection, it is strange that the input to STRling are reads that are aligned to known STR regions, or are unmapped reads, from a BAM or CRAM file. Most repeat detection algorithms (or even some SV detection algorithms) suggest a different set of parameters for alignment, to improve the quality of STR discovery and quantification. The impact of alignment software (and specific sets of parameters) need to be taken into account in the analysis.

Our approach is designed to work with the most common alignment strategy (BWA MEM) to avoid the massive computational overhead required to perform complete realignment (as has been done in the papers describing STRetch and lobSTR). BWA MEM has been found to perform well at short STR loci (e.g. <https://www.nature.com/articles/nmeth.4267>). We recognize that the aligner will likely mis-align many reads containing substantial STR content. Instead of re-aligning all reads we “fix” the alignment of these STR reads within STRling before proceeding. When choosing which reads to assess for STR content we consider reads aligned to known STR regions, unaligned reads, and reads with mismatches/indels/softclipping. We have found this to be a sensible heuristic to gather all informative STR reads without unnecessarily performing k-mer counting on the entire BAM/CRAM file.

4. The authors used a set of filters to drop down candidate regions significantly. On average, the 134 subjects tested had a median of 9 significant STR expansions each. It is difficult to judge the reliability of such calls without experimental validation, but one easy thing to do is to assess similar number of subjects in the same sequencing batch who are not carriers or not affected, and see how the results differ.

As all the individuals in the same batch as the affected individuals were either affected or the immediate family member of an affected individual (many of whom were carriers), we looked to the 1000 Genomes to answer this question. We randomly sampled cohorts of various sizes and performed joint calling and outlier testing. In Supplementary Figure 4B we report the number of outliers per individual with varying filtering strategies. Using a cohort of 100 individuals, a similar size to the true positive cohort, we applied the recommended filters, namely: outliers with 2-6 bp repeat units that do no overlap segmental duplications, LCRs, centromeres or telomeres. We found an average of 15.7 outliers per individual (range: 7-14).

We randomly sampled individuals from the 1000 Genomes Project, performed STRling joint calling on each subset, and reported the number of significant outliers per individual. All outliers on canonical chromosomes chr1-22, X, and Y, outliers at 2-6bp repeat unit loci, and outliers at 2-6bp repeat unit loci excluding those overlapping LCRs, segmental duplications, centromeres, or telomeres.

5. *The claim about "PCR is the current standard method for STR disease diagnostics" is not correct. It may be the case for certain loci in certain diseases (such as many polyQ diseases where the expanded repeats are merely a few hundred base pairs), but definitely cannot be extrapolated to all repeat expansion disorders. Some can benefit from repeat primed PCR (which is different from PCR), with pulse field, yet others depend on Southern blot or capillary electrophoresis, and some others depend on highly specialized enzymes for PCR, and some depends on other more modern method such as molecular combing. It is all just very much disease dependent and loci dependent. The diseases assays in this manuscript (for example, Figure 3) only represent a portion of the diseases where repeat plays a role. This is something not made explicit and give readers a false impression that NGS can actually solve the problem of repeat expansions, which is not the case; considering that so many similar software tools have already been published yet the true clinical problem has not actually been solved yet, it highlight the challenge in even using NGS to study these diseases or to find novel STRs. Having said that, I also acknowledge that there are specific application scenarios (one example is forensic diagnosis) where the labs began to design specializes assays and then use NGS to do STR typing, but then again it is not the same as whole-genome NGS.*

Thank you for the suggestion. We have made the following additions to the text to emphasize the range of laboratory techniques used in PCR analysis:

Introduction:

Genotyping individual STR expansions using established laboratory techniques such as conventional or repeat-primed Polymerase Chain Reaction (PCR), or Southern blot, capillary or pulse-field gel electrophoresis is expensive and time-consuming, requiring locus-specific assay development. Some phenotypes may be caused by one of several SNV and STR variants, such that even disease panels may still miss causal STR expansions [13]. Increasingly, researchers are moving to genome sequencing, which is often more economical and may yield a faster diagnosis [14].

Results:

For 103 of the subjects with known STR disease, we also had orthogonal allele size estimates from repeat-primed PCR. It should be noted that while PCR-based methods are frequently used for STR disease diagnostics, the accuracy of PCR allele size estimates can suffer from stutter and allelic dropout [29,30].

Our validations/size estimates in affected individuals were done by traditional, long-range, or repeat-primed PCR, in conjunction with capillary electrophoresis. In many cases these were done as part of a clinical or research diagnostic panel.

6. I had difficulty with the entire section of "Long reads enable estimates of STRling's false discovery rate". I would have expected (when reading earlier portion of the manuscript) that if PacBio HiFi is used, the authors would be able to either identify known STRs from patients, or find novel (not present in the reference genome) STRs that may lead to human diseases. Yet the entire section is about "false discovery rate": not that it is not important, but that this is not one typical reader expect. Many of disease-causal repeat expansions that are discovered in the past 3 years (for example, CANVAS and FAME if I remember correctly though I could be wrong) are actually found by long reads, not short reads, and they would have served as perfect examples of how STRling can possibly work on long read. Yet the whole section is about healthy individuals such as Ashkenazim trio and I do not really see the real point of this section here compared to the real clinical problem that a method such as STRling is trying to solve.

STRling is a method to detect STR expansions from short reads. It does not work on long reads. It relies on paired-end information and the assumption that pathogenic alleles will typically be larger than the reads. It also assumes high base-pair sequencing accuracy. We have plans to adapt the underlying STRling k-mer counting strategy to develop a method capable of calling STRs in long reads, however, this will be the subject of a future publication.

We have not been able to find individuals with known disease STR expansions who have been sequenced with both Illumina WGS and a long-read technology. Therefore we chose to use previously conducted PCR as an orthogonal method to verify the pathogenic STR expansions in individuals with known STR disease to assess STRling's sensitivity. The purpose of the long-read analysis was to determine if STRling calls at likely non-pathogenic loci were true or false positives. For this reason, we used unaffected individuals that have

been sequenced with both long and short-read sequencing as an additional strategy to estimate STRling's false discovery rate.

7. There is a general lack of comparison to existing computational tools. This field is flooded with bioinformatics methods and it would be ideal to do more thorough comparison to show what are the advantages. There are some comparisons such as ExpansionHunter Denovo but certainly not enough, and there are sections such as "algorithm resource requirements" where there is basically no comparison so a reader would have no real idea whether the time/memory requirement improves over existing tools or is excessive. In fact, I would argue that requirement about the joint merge stage (29.04 GB) and the joint outlier stage (27.05 GB) sound quite resource-intensive and not really applicable to typical labs unless dedicated servers are used.

While there are several methods for genotyping STRs at known reference loci, only ExpansionHunter Denovo (EHdn) and STRling are capable of discovering novel expansions. We have added resource usage for EHdn run on the same 260 WGS that were used for STRling on the same compute cluster. This has been added to Supplementary Figure 6. In summary, EHdn profile stage typically takes less than 40 minutes and uses a little under 1 GB of memory. This is slightly lower, but still comparable to the STRling extract stage which takes about 60 mins and uses a little over 1 GB of memory. Where the two methods differ is the merge and outlier stages. For merge, STRling uses 29 GB of memory and 56 minutes, compared to less than a minute and 0.4 GB. For the outlier stage, STRling uses 27 GB of memory and takes 22 minutes. EHdn takes 34 minutes to run the outlier stage, but much less memory, at 0.3 GB.

While STRling does have higher resource requirements than EHdn, they are still within the scale of resources commonly used for human WGS variant calling, for example, GATK. Another key consideration when comparing STRling resource requirements to EHdn, is that STRling is doing substantially more work. STRling uses soft-clipped reads to provide base-pair accurate loci, while EHdn only uses paired-end information. This dramatically increases the number of reads that STRling must look at, but also increases its position accuracy. STRling does allele size estimation, while EHdn does not. Again, this requires looking at more reads. We trade-off a modest increase in resource requirements to gain additional allele size information and position accuracy.

We have added EHdn resource usage as Supplementary Figure 6 and commented on it in the main text.

ExpansionHunter Denovo resource usage. A EHdn outlier workflow was executed on the same 260 WGS from the 1000 Genomes Project. **A**: Time and **B**: memory usage.

8. There are many places that can be considered as over statement. Take an example in Abstract: "It is the first method to resolve the position of novel STR expansions to base pair accuracy", it can certainly be challenged by other authors whether short or long read or pseudo-long read were used in previous publications, there are many previous examples of finding novel STR expansions with base pair resolution which is exactly why so many repeat expansion diseases are actually discovered over the past few years.

We agree that this was an overstatement and should be qualified. It has been removed from the abstract for brevity. A related statement does remain in the conclusion, specifically: "In

contrast to previous computational methods to detect novel STRs, STRling is capable of defining the locus boundaries to base-pair accuracy.”

There is one prior computational method to discover novel STRs in short reads, ExpansionHunter Denovo, and it does not find them with bp accuracy (Figure 3). We agree that previous studies have discovered loci with base-pair accuracy using various long and short-read sequencing methods, and other techniques. However, strategies based on short reads typically involved some manual intervention.

Second round of review

Reviewer 1

The authors have adequately addressed all my concerns. It is worth noting that, in addition to autism, intelligence and depression, tandem repeat expansions have also been associated with schizophrenia (<https://doi.org/10.1038/s41380-022-01575-x>). The authors may want to add this study to their references.

Reviewer 2

The authors have addressed my previous comments adequately with additional analysis and clarification in the revised manuscript.